

# Remote sensing of aerosol water fraction, dry size distribution and soluble fraction using multi-angle, multi-spectral polarimetry

Bastiaan van Diedenhoven[1], Otto P. Hasekamp[1], Brian Cairns[2], Gregory L. Schuster[3], Snorre Stamnes[3], Michael Shook[3], and Luke Ziemba[3]

[1]SRON Netherlands Institute for Space Studies, Leiden, the Netherlands
[2]NASA Goddard Institute for Space Studies, New York, New York, USA
[3]NASA Langley Research Center, Hampton, Virginia, USA.

**Correspondence:** Bastiaan van Diedenhoven (b.van.diedenhoven@sron.nl)

**Abstract.** A framework to infer volume water fraction, soluble fraction and dry size distributions of fine mode aerosol from multi-angle, multi-spectral polarimetry retrievals of column-averaged ambient aerosol properties is presented. The method is applied to observations of the Research Scanning Polarimeter (RSP) obtained during two NASA aircraft campaigns, namely the Aerosol Cloud meTeorology Interactions oVer the western ATlantic Experiment (ACTIVATE) and the Cloud, Aerosol, and
Monsoon Processes-Philippines Experiment (CAMP²Ex). All aerosol retrievals are statistically evaluated using in situ data. Volume water fraction is inferred from the retrieved ambient real part of the refractive index, assuming a dry refractive index of 1.54 and by applying a volume mixing rule to obtain the effective ambient refractive index. The uncertainties in inferred volume water fraction resulting from this simplified model are discussed and estimated to be lower than 0.2 and decreasing with increasing volume water fraction. The daily mean retrieved volume water fractions correlate well with the in situ values
with a mean absolute difference of 0.09. Polarimeter-retrieved ambient effective radius for daily data is shown to increase as a function of volume water fraction as expected. Furthermore, the effective variance of the size distributions also increases with increasing effective radius, which we show is consistent with an external mixture of soluble and insoluble aerosol. The relative variations of effective radius and variance over an observation period are then used to estimate the soluble fraction of the aerosol. Daily results of soluble fraction correlate well with in situ observed sulfate mass fraction with a correlation
coefficient of 0.79. Subsequently, inferred water and soluble fractions are used to derive dry fine-mode size distributions from their ambient counterparts. While dry effective radii obtained in situ and from RSP show similar ranges, in situ values are generally substantially smaller during the ACTIVATE deployments, which may be due to biases in RSP retrievals or in the in situ observations, or both. Both RSP and in situ observations indicate the dominance of aerosol with low hygroscopicity during the ACTIVATE and CAMP²Ex campaigns. Furthermore, RSP indicates a high degree of external mixing of particles
with low and high hygroscopicity. These retrievals of fine mode water volume fraction and soluble fraction may be used for the evaluation of water uptake in atmospheric models. Furthermore, the framework allows to estimate the variation in the concentration of fine-mode aerosol larger than a specific dry radius limit, which can be used as a proxy for the variation in cloud condensation nucleus concentrations. This framework may be applied to multi-angle, multi-spectral satellite data expected to be available in the near future.



# 1 Introduction

Terrestrial aerosol is a complex mixture of liquid, solid, or mixed-phase particles emitted by natural and anthropogenic sources either directly to the atmosphere or produced in the atmosphere through complex physical and chemical pathways from precursor gases (Riemer et al., 2019). Aerosols affect Earth's climate as they scatter and absorb radiation and act as condensation
nuclei for cloud droplets and ice crystals. Furthermore, aerosols affect air quality and health.

Substantial changes in anthropogenic aerosol emissions in the industrial age occurred (Bauer et al., 2020). The overall increase in aerosol emissions have lead to a general cooling of Earth's atmosphere, compensating part of the temperature increase imposed by anthropogenic greenhouse gas emissions (Samset et al., 2018; Bauer et al., 2020). Reductions in global aerosol emissions are expected in the next decades, improving air quality, but also likely providing a net positive climate
forcing. Prevailing uncertainties in the effective radiative forcing from anthropogenic aerosol emissions continues to hamper improving the accuracy of estimates of global climate sensitivity to changes in greenhouse gas concentrations (IPCC, 2021). A better understanding of the physics and chemistry of terrestrial aerosols and an improved representation of aerosols and their direct and indirect radiative effects in climate models are essential for reducing the uncertainties in modeled climate sensitivity.

Both the climate and air quality effects of aerosols depend on their concentrations, size, shape, composition, hygroscopicity
and mixing state, as well as on their geographical location and meteorological environment. Models generally define a number of aerosol types or modes with fixed properties, such as composition and hygroscopicity, and diagnose their evolution with respect to, e.g., concentrations, size and internal and/or external mixing state (e.g., Bauer et al., 2008; Zhang et al., 2012). Modeled aerosol composition and especially water uptake is highly variable among models (Textor et al., 2006). Satellite remote sensing products are essential to evaluate these global aerosol simulations and the direct and indirect effects, although
such evaluations are hampered by the limited information content of most traditional satellite observations generally providing only aerosol spectral optical depth. For example, using variations in spectral optical depth to estimate variations in cloud condensations nuclei (CCN) concentrations may lead to substantial biases in assessments of modeled aerosol-cloud interactions (Hasekamp et al., 2019). Furthermore, many combinations of aerosol loading, size, hygroscopicity and atmospheric humidity may lead to the same aerosol optical depth and thus biases in these properties may compensate each other (Bian et al., 2009).
Multi-angle, multi-spectral polarimetric satellite observations provide extended information content for remote sensing of aerosol properties, yielding information about size, complex refractive index and height (Mishchenko et al., 2004; Hasekamp and Landgraf, 2007; Wu et al., 2016; Xu et al., 2017; Dubovik et al., 2019), which can be used to determine and to better constrain aerosol types (Kacenelenbogen et al., 2022), emissions and evolution (e.g., Chen et al., 2019; Tsikerdekis et al., 2021) and aerosol-cloud interactions (Hasekamp et al., 2019).
Of particular interest is to constrain the amount of water in the aerosol, as this affects the aerosol optical depth, size and absorption. Quantification of water fraction and mixing state allows to estimate dry size distribution based on their ambient counterpart that is inferred, e.g., using polarimetry. Dry size is one of the main factor determining whether an aerosol particle



is an effective CCN at a given supersaturation or not (Dusek et al., 2006; Crosbie et al., 2015). Hence, the variation in the concentration of fine-mode aerosol larger than a specific dry radius limit may be used as a proxy for variation in CCN concentrations. This concept was applied to multi-angle, multi-spectral polarimetry retrievals by Hasekamp et al. (2019). However, in their implementation ambient size distributions were used in lieu of dry size distribution retrievals, which may lead to a bias in estimated variation in CCN concentrations related to water uptake.

Of additional interest is quantification of the aerosol mixing state. Zheng et al. (2021) found a high degree of external mixing of hygroscopic (soluble) and nonhygroscopic (insoluble) aerosol components that is seasonally varying. Consequently, humidification of the aerosol leads to an unequal distribution of the water of the aerosol population. Ching et al. (2017) showed that an unrealistic assumption of 100% internally mixed aerosol may lead to errors in estimated CCN concentrations from model simulations that may exceed 100%, depending on the true mixing state. The degree to which an aerosol is externally mixed also depends on the considered vertical and horizontal spatial scales and the inhomogeneity of aerosol properties on those scales.

Here, we provide a framework for deriving water volume fraction, dry size distributions and soluble fraction of fine mode aerosol from the ambient aerosol properties retrieved by multi-angle, multi-spectral polarimeters. Fine mode water volume fraction is inferred using the retrieved real part of the refractive index. Since the refractive index of water is lower than that of dry aerosol, the refractive index decreases as the total water volume fraction in the aerosol increases. This behavior has been used previously to estimate the aerosol volume fraction of water, in addition to general composition, from ground-based sun-photometer observations (Schuster et al., 2009; Wang et al., 2013; van Beelen et al., 2014; Zhang et al., 2020). Here we propose a simplified approach focusing on water volume fraction only. Furthermore, we show that the soluble fraction can be estimated from a set of aerosol size distributions retrievals collected over a time period (or within a region) during which a substantial variation in volume water fraction occurs. Finally, the water and soluble fractions are used to derive effective radius and variance of the dry size distributions.

The presented method is applied to observations of the airborne Research Scanning Polarimeter. We evaluate the results using in situ observations that are made during the same campaign flights. The method and data are described in sections 2 and 3 after which results are presented in section 4. The assumptions and related uncertainties are discussed in section 5 after which we conclude the paper in section 6

## 2 Definitions and method

### 2.1 Aerosol water uptake: humidified soluble aerosols

The number size distribution $N_a(r)$ of aerosols in terms of their spherical radius $r$ is represented by a log-normal distribution of the form (Seinfeld and Pandis, 1998; Stamnes et al., 2018)

$$N_a(r) = \frac{N_{tot}}{r\sqrt{2\pi}\sigma_g}\exp\left[-\left(\frac{\ln r - \ln \bar{r}_g}{\sqrt{2}\sigma_g}\right)^2\right], \tag{1}$$





where $N_{tot}$ is the total aerosol number concentration, $\overline{r}_g$ and $\sigma_g$ are the geometric mean radius and geometric standard deviations, respectively. For a log-normal distribution, the geometric mean radius also corresponds to the distribution's median. Other commonly used parameters to indicate the characteristic size and width of a size distribution are the effective radius $r_e$ and effective variance $v_e$, formally defined as respectively

$$r_e = \frac{\langle r^3 \rangle}{\langle r^2 \rangle} \tag{2}$$

and

$$v_e = \frac{\langle r^4 \rangle \langle r^2 \rangle}{\langle r^3 \rangle^2} - 1, \tag{3}$$

where $\langle r^x \rangle$ represents the $x$th moment of the size distribution (Hansen and Travis, 1974). For a log-normal distribution (Eq 1), $r_e$ and $v_e$ are related to $\overline{r}_g$ and $\sigma_g$ by

$$r_e = \overline{r}_g \, \exp\left(-\frac{5}{2}\,\sigma_g^2\right) \tag{4}$$

and

$$v_e = \exp(\sigma_g^2) - 1. \tag{5}$$

Upon water uptake, a soluble aerosol will be diluted resulting in an increase of its radius and a change in its refractive index. For a particle with dry radius $r_{dry}$, the wet radius $r_{RH}$ it will obtain upon exposure to relative humidity $RH$ (expressed here as a fraction) is given by "kappa-Köhler" theory (Petters and Kreidenweis, 2007), namely by

$$RH = \frac{r_{RH}^3 - r_{dry}^3}{r_{RH}^3 - r_{dry}^3(1-\kappa)} \exp\left(\frac{2\sigma_s M_w}{RT\rho_w r_{RH}}\right), \tag{6}$$

where $\sigma_s$ is the surface tension of water at the particle-to-air interface, $R$ is the universal gas constant, $T$ is temperature and $M_w$ and $\rho_w$ are the molecular weight and density of water, respectively. Furthermore, $\kappa$ is a variable to parameterize the hygroscopicity of the aerosol. Increasing hygroscopicity is parameterized by increasing $\kappa$, while for completely insoluble particles $\kappa \equiv 0$. The exponential term in Eq.6 represents the effects of curvature of the drop on its growth and is commonly referred to as the Kelvin term. In practice, the Kelvin term is often ignored as its effect is only substantial for very low $\kappa$ and/or small radii. Setting the Kelvin term to unity and defining a particle volume growth factor

$$g_{V,sol} = \frac{r_{RH}^3}{r_{dry}^3}, \tag{7}$$

we obtain (cf. Brock et al., 2016a)

$$g_{V,sol}(RH) \approx 1 + \kappa \, \frac{RH}{1-RH}. \tag{8}$$

Since the Kelvin term is ignored, the volume growth factor does not depend on size and Eq. 8 also represents the bulk volume growth factor of the size-integrated wet particle volume. Hence, the geometric mean radius of the size distribution of the humidified aerosol is then approximated by

$$\overline{r}_{g,wet}(RH) = \overline{r}_{g,dry} \, g_{V,sol}^{1/3}(RH). \tag{9}$$




The geometric standard deviation does not change with water uptake if the Kelvin term is ignored. Hence, the effective radius also scales as

$$r_{e,wet}(RH) = r_{e,dry} \, g_{V,sol}^{1/3}(RH). \tag{10}$$

Furthermore, the volume fraction of water in an humidified soluble aerosol $f_w$ is given by

$$f_w(RH) = \frac{g_{V,sol}(RH) - 1}{g_{V,sol}(RH)}. \tag{11}$$

equivalent to

$$g_{V,sol}(RH) = \frac{1}{1 - f_w(RH)}. \tag{12}$$

For an arbitrary dry effective radius of 0.15 $\mu$m, the increase of $r_{e,wet}$ as a function of $f_w$ is shown in Fig. 1 (black line). In case of humidified soluble aerosol and known aerosol volume water fraction $f_w$, the dry geometric and effective radii can be estimated from their wet equivalents, using Eq. 12 and Eq. 9 and 10, respectively.

### 2.2 Aerosol water uptake: external mixtures of humidified aerosol and insoluble aerosol

Aerosol populations may also be represented as external mixtures of soluble (with $\kappa_{sol} > 0$) and insoluble ($\kappa_{insol} \equiv 0$) particles (Heintzenberg et al., 2001; McFiggans et al., 2006; Swietlicki et al., 2008; Wex et al., 2010; Holmgren et al., 2014; Riemer et al., 2019; Kim et al., 2020). As evident from Eq. 8, insoluble aerosol do not grow upon exposure to humidity (i.e., $g_{V,insol} = 1$). For such aerosol populations, the increase of aerosol size upon exposure to a given relative humidity depends on the fractional contribution by soluble particles to the total population in addition to their hygroscopicity. Furthermore, since the insoluble particles are not growing with increasing relative humidity, the total aerosol size distribution widens upon humidification depending on the fraction of soluble particles. Here, we describe the relationship between the size distribution parameters, water fraction and the fraction of soluble particles.

Defining $f_{sol}$ as the soluble volume fraction of a dry external mixture of soluble and insoluble particles, the mixture growth factor $g_{V,mix}$ is given by (omitting the $RH$ dependencies here and in the rest of this section)

$$g_{V,mix} = f_{sol} \, g_{V,sol} + (1 - f_{sol}) \tag{13}$$

equivalent to

$$g_{V,sol} = \frac{g_{V,mix} + f_{sol} - 1}{f_{sol}}. \tag{14}$$

The water volume fraction of the mixture is related to the growth factor as

$$g_{V,mix} = \frac{1}{1 - f_w}, \tag{15}$$

where $f_w$ represents the volume water fraction of the total aerosol population, i.e. including soluble and insoluble particles.





The size distribution of the mixture of humidified soluble aerosol and insoluble aerosol can be represented by a bi-modal log-normal distribution of which the geometric mean and standard deviations are given by respectively

$$\ln(\overline{r}_{g,mix}) = f_{sol} \ln(\overline{r}_{g,wet}) + (1 - f_{sol}) \ln(\overline{r}_{g,insol}) \tag{16}$$

and

$$\sigma^2_{mix} = [\ln(\overline{r}_{g,insol}) - \ln(\overline{r}_{g,wet})]^2 \, f_{sol} \, (1 - f_{sol}) + f_{sol}\sigma^2_{wet} + (1 - f_{sol})\sigma^2_{insol}. \tag{17}$$

We make the assumption that the size distributions of the insoluble component and the dry soluble component of the mixture are the same, i.e. $\overline{r}_{g,insol} = \overline{r}_{g,dry}$ and $\sigma_{insol} = \sigma_{dry}$. Then, combining Eqs. 9 with respectively 16 and 17 leads to

$$\overline{r}_{g,mix} = \overline{r}_{g,dry} \, g_{V,sol}^{f_{sol}/3} \tag{18}$$

and

$$\sigma^2_{mix} = \sigma^2_{dry} + \frac{1}{9}\ln(g_{V,sol})^2 \, f_{sol} \, (1 - f_{sol}). \tag{19}$$

When approximating the resulting size distribution of the mixture as a single log-normal distribution, its effective radius and variance can be approximated by respectively

$$r_{e,mix} = r_{e,dry} \, g_{V,sol}^{f_{sol}/3} \, \exp\left[\frac{5}{18}\ln(g_{V,sol})^2 \, f_{sol} \, (1 - f_{sol})\right] \tag{20}$$

and

$$v_{e,mix} = (v_{e,dry} + 1) \, \exp\left[\frac{1}{9}\ln(g_{V,sol})^2 \, f_{sol} \, (1 - f_{sol})\right] - 1. \tag{21}$$

From Eqs. 20, 21, 14 and 15 it is clear that effective radius generally increases with $f_w$ when $f_{sol} > 0$, while the effective variance also increases with $f_w$ when $0 < f_{sol} < 1$. Note that, under the assumptions laid out in the text above, the relationships between $f_w$ and $r_{e,mix}$ and $v_{e,mix} + 1$ only depend on $f_{sol}$ and not on the size distributions themselves or on $\kappa$.

        The variation of $r_{e,mix}$ and $v_{e,mix}$ with $f_w$ for various values of $f_{sol}$ are shown in Figs. 1a and 1b, respectively, for arbitrary dry values of 0.15 $\mu$m and 0.15, respectively. Note that these curves are derived not by using Eqs. 20 and 21, but by calculating
the geometric mean and standard deviation of a bi-modal size distribution resulting from an external mixture of humidified soluble and insoluble aerosol and applying Eq. 4 and 5 to obtain corresponding effective radii and variances. However, applying Eq. 20 and 21 directly yields similar results. Fig. 1a shows that, for $1 \geq f_{sol} > \sim 0.15$, the increase of $r_{e,mix}$ with $f_w$ is generally steeper than for $f_{sol} = 1$, with the steepest increase at $f_{sol} \sim 0.5$, while for $f_{sol} < \sim 0.15$ the relationship between $r_{e,mix}$ and $f_w$ is shallower. The increase of $v_{e,mix}$ as a function of $f_w$ is shown in Fig. 1b. As discussed in section 2.1, the effective variance
does not change with water fraction if $f_{sol} = 1$. Generally the slope of $v_{e,mix}$ with respect to $f_w$ increases with decreasing $f_{sol}$ when $f_{sol} > \sim 0.15$. For $f_{sol} < \sim 0.15$, the relationship between $v_{e,mix}$ and $f_w$ is shallower and notably different than for higher soluble fractions. From Eqs. 20 and 21 we can conclude that the slope of $r_{e,mix}$ and $v_{e,mix} + 1$, both relative to the dry



values only depends on $f_{sol}$. This is shown in Fig. 1c where $v_{e,mix} + 1$ is plotted against $r_{e,mix}$, both relative to the dry values, revealing approximately linear relationships with a systematically increasing slope with decreasing $f_{sol}$. As will be shown in
section 4.2, the slope of retrieved $r_e$ with respect to retrieved $v_e + 1$ within a region or time period may be used to infer $f_{sol}$.

In case of an external mixture of humidified soluble aerosol and dry insoluble aerosols, the dry effective radius and variance may be retrieved from their ambient values for known aerosol volume water fraction $f_w$ *and* known $f_{sol}$ using Eqs. 20 and 21.

### 2.3 Refractive indices of aqueous solutions of aerosol and their external mixtures with insoluble aerosol

To obtain the refractive index of an aqueous solution $n_{wet}$ of aerosol with known dry refractive index $n_{dry}$, several mixing
rules are commonly used, of which the Lorentz-Lorenz and molar refractions mixing rules are generally considered the most fundamental (Brocos et al., 2003; Liu and Daum, 2008). However, the simplest approach that is perhaps most widely used in the atmospheric science community is to estimate $n_{wet}$ by a volume-weighted average of $n_{dry}$ and that of water $n_{water}$, i.e.

$$n_{wet} = f_w \, n_{water} + (1 - f_w) \, n_{dry}. \tag{22}$$

In case of ideal mixtures, i.e. when the volume (or density) of a mixture is equal to the sum of the volumes (or densities) before
mixing, differences between $n_{wet}$ calculated using volume-averaging (Eq. 22) and Lorentz-Lorenz or molar refraction mixing may be considered negligible (Brocos et al., 2003). In case of substantial deviations from ideality, however, as observed for pure inorganic salts such as ammonium sulfates and sodium chloride, Eq. 22 substantially overestimates $n_{wet}$ for solutions with water fractions below about 50% (Tang and Munkelwitz, 1991; Schuster et al., 2009; Erlick et al., 2011, see also supplementary material). Furthermore, salts effloresce at low humidities leading to maximum refractive indices in their aqueous state that are
substantially lower than their refractive index in the dry state (Schuster et al., 2009). In the case of aqueous solutions of soluble organic aerosol, however, refractive indices are well approximated by Eq. 22 (Lienhard et al., 2012; Cai et al., 2016, see also supplementary material). Moreover, refractive indices of ternary aqueous mixtures containing both organics and inorganic salts at a 1:1 molar ratio are also well approximated by Eq. 22 over the full range of $f_w$, as also demonstrated in the supplement based on the data of Lienhard et al. (2012). Furthermore, effloresence of the salt seems to be suppressed by the organics in
the solution (Lienhard et al., 2012; Jing et al., 2016; Wu et al., 2020). Since aerosol particles are generally internally mixed with many components (Riemer et al., 2019), densities are not known and accounting for density changes when estimating refractive indices of the mixtures is practically impossible. However, organics generally make up 20% to 90% of the particle mass (Jimenez et al., 2009), which may be considered sufficient for the volume mixing rule of Eq. 22 to be generally applicable (cf. Stokes and Robinson, 1966).

In case of externally mixed aerosol, the retrieved refractive index inferred by remote sensing or in situ observations may be considered an effective value that is radiatively equivalent to that of the externally mixed aerosol. Here, we use the volume mixing rule also to estimate the effective refractive index of externally mixed aerosols based on the refractive indices of individual components in the mixture. Using simulations presented in the supplement, we conclude that a volume mixing rule generally yields the effective refractive index of an external mixture to within about 0.02. Hence, for mixtures of soluble (with



$\kappa_{sol} > 0$) and insoluble ($\kappa_{insol} \equiv 0$) aerosol, the refractive index of the mixture $n_{mix}$ is then approximated by

$$n_{mix} = f_w \, n_{water} + (1 - f_w) \left[ f_{sol} \, n_{dry,sol} + (1 - f_{sol}) \, n_{insol} \right], \tag{23}$$

where $f_w$ is the volume fraction of water within the total aerosol population and $n_{dry,sol}$ and $n_{insol}$ are the refractive indices of the dry soluble and insoluble aerosol particles, respectively. If $n_{dry}$ is taken as the weighted average of $n_{dry,sol}$ and $n_{insol}$, Eq. 23 is equivalent to Eq. 22. Hence, for our purpose of inferring volume water fraction from the retrieved effective refractive

indices, the mixing state of the aerosol is generally irrelevant.

In case of humidified aerosol, its aerosol volume water fraction $f_w$ may be estimated from its refractive index $n_{wet}$ using Eq. 22 and under assumptions of $n_{dry}$ and $n_{water}$. At a wavelength of 555 nm, $n_{water}$ is 1.3337 (Segelstein, 1981). For a wide variety of aerosol types, including marine, biogenic, urban, background and tropospheric aerosols and those associated with of biomass burning from agricultural and wild fires, many studies (Levoni et al., 1997; Sorooshian et al., 2008; Shingler

et al., 2016a; Brock et al., 2016b; Aldhaif et al., 2018; Espinosa et al., 2019; Bian et al., 2020) found that effective real parts of the refractive index for dry fine-mode aerosol are about 1.52–1.54 on average. While observed ranges may extend from about 1.42 to 1.60, the standard deviations (or interquartile ranges) of the observations are generally small at about 0.02 (or 0.04). The refractive index of an aerosol depends on its chemical composition and mixing state, which are in turn determined by its emission source, age and history of environmental conditions (Riemer et al., 2019). However, relationships between these

factors and dry refractive indices are highly uncertain. For example, dry refractive indices of organic aerosol have been reported to increase or decrease with chemical aging (Mack et al., 2010; Moise et al., 2015; Li et al., 2017; He et al., 2018; Aldhaif et al., 2018). Individual particles within an aerosol are complex mixtures of different chemical species (Riemer et al., 2019). Lang-Yona et al. (2010) showed that dry refractive indices of such internally mixed aerosol are generally well approximated by a weighted average of their components using a volume mixing rule. Since refractive indices of organic compounds, as well as

of many common inorganics, are generally within the range of 1.48–1.60 (e.g, Aldhaif et al., 2018), it may not be surprising that the observed mean effective refractive indices of fine mode aerosol in various airmasses are generally close to 1.54.

While systematic relationships of dry refractive index with source, age and environmental conditions may be hard to establish, a systematic decrease of refractive index with increased water fraction in the aerosol may generally be assumed, since the refractive index of water is generally smaller than that of dry aerosol. As argued above, for chemically heterogeneous aerosol

that is internally and externally mixed, a linear decrease in refractive index with increased water volume fraction as represented by Eqs. 22 and 23 may be assumed. Figure 2 shows the refractive index of a humidified aerosol as a function of the volume water fraction, as calculated by Eq. 22, under the assumption that $n_{dry} = 1.54$. This relationship is the basis of the volume water fraction retrievals from the observed ambient fine mode refractive index presented in this paper. The ranges of refractive index of the aqueous mixture when assuming the dry refractive index is smaller/greater than 1.54 by 0.02 and 0.04 are also shown

in Fig. 2 by the blue and grey shading, respectively. Hence, the uncertainty of the volume water fraction inferred from a given refractive index decreases with water fraction and has a maximum of about 0.18 and 0.32 when assuming the uncertainty in dry refractive index to be ±0.02 and ±0.04, respectively. Under the consideration of the assumptions and uncertainties discussed above, uncertainties in retrieved water fraction from an observed ambient refractive index are likely within those represented



by the blue range in Fig. 2 for most cases and regions, while the greater uncertainties represented by the grey range may occur
for cases and regions with particularly low or high dry refractive indices.

## 3   Data

### 3.1   Campaigns

The data used in this paper were collected during the NASA the Aerosol Cloud meTeorology Interactions oVer the western
ATlantic Experiment (ACTIVATE) aircraft campaign (Sorooshian et al., 2019; Corral et al., 2021) and the NASA Cloud,
Aerosol, and Monsoon Processes-Philippines Experiment (CAMP²Ex) aircraft campaign (Hilario et al., 2021; Reid et al.,
2022).

The ACTIVATE is a five-year long campaign based at NASA Langley Research Center in Hampton, VA on the US East
Coast. The overarching goal of the campaign is to robustly characterize aerosol-cloud-meteorology interactions using exten-
sive, systematic, and simultaneous in situ and remote sensing airborne measurements with two aircraft and a hierarchy of
models. Here we use data from the first two deployments of the campaign in the winter and summer of 2020. Remote sensing
instruments, including the RSP, were mounted on a King Air aircraft, while in situ observations were made using a HU-25
Falcon aircraft. Most flights were coordinated so that the remote sensing and in situ were obtained close in time and in the
same area. Only data over ocean are used here.

The CAMP²Ex campaign was based at Clark airport on Luzon, the Philippines, from 24 August to 5 October 2019. It was
designed to study the covariability and mutual influences of aerosol, clouds, chemistry, meteorology, convection and radiation.
All remote sensing and in situ observations used in the paper were made with instruments mounted on the P3-B aircraft which
flew at altitudes near the surface to about 8 km. Only data over ocean are used here.

The dates and times at which data was collected are listed in the supplement in Tables S2 and S3.

### 3.2   Remote sensing observations

Aerosol properties are retrieved from multi-angle, multi-spectral total and polarized radiances using the Microphysical Aerosol
Properties from Polarimetry (MAPP) algorithm described by Stamnes et al. (2018). This algorithm is based on an optimal
estimation approach and retrieves the aerosol optical depth at 555 nm, effective radius and effective variance in both a fine and
coarse aerosol size mode, in addition to the complex refractive index at 555 nm and layer height of the fine mode. The coarse
mode is assumed to have the refractive index of hydrated sea salt and is homogeneously mixed below 0.5 km. The surface
reflectance is modeled using a bio-optical ocean model in terms of a chlorophyll concentration and windspeed, which are also
retrieved alongside the aerosol parameters.

The fine mode total number concentrations $N_{tot,f}$ are subsequently derived using the approximation (Schlosser et al., 2022)

$$N_{tot,f} = \frac{\tau_f}{\sigma_f \, \Delta z_f},$$

(24)



where $\tau_f$ and $\Delta z_f$ are the retrieved fine mode aerosol optical depth and layer height and $\sigma_f$ is the extinction cross section, which in turn is calculated using the retrieved fine mode size distribution and Mie theory.

The MAPP algorithm is applied to data from the Research Scanning Polarimeter (RSP, Cairns et al., 1999), which measures the I, Q, and U Stokes parameters at nine wavelengths in the visible and shortwave infrared. RSP scans along track at a rate of about 0.86 seconds, during which observations at more than 100 viewing angles are collected. The latitude and longitude of

each view projected to the surface is determined and the data is rearranged so that multi-angle views at consecutive 'pixels' on the surface are obtained. To improve the RSP data signal and reduce the computational effort of retrieval processing, data for 20 consecutive pixels are averaged together and used as input to the MAPP retrieval algorithm. To avoid clouds, generally data closer than 24 scans to clouds are removed. For ACTIVATE this is reduced to 12 scans to increase the number of data points. Note that contamination by clouds below the aircraft as well as substantial cirrus clouds above the aircraft is effectively filtered

out by the goodness-of-fit tests applied in the aerosol retrieval algorithm (Stap et al., 2015; Stamnes et al., 2018).

### 3.3    In situ observations

In situ aerosol observations used here are obtained with the Langley Aerosol Research Group Experiment (LARGE) instrument package or both ACTIVATE and CAMP2Ex datasets. All aerosol measurements are made from a forward-facing, shrouded, solid diffuser inlet that efficiently samples particles with aerodynamic diameter less than 5.0 $\mu$m (McNaughton et al., 2007).

Observations in cloud are excluded by requiring that the liquid water content observed by the (Fast) Cloud Droplet Probe (Stratton Park Engineering Company, Boulder, CO, USA) is below 0.02 g/cm$^3$ and that the ambient relative humidity obtained by the Diode Laser Hygrometer (DLH Podolske et al., 2003) is below 100%.

In general, aerosol particles are dried before their properties are measured by in situ probes. However, their hygroscopicity is estimated by using a nephelometer (model 3563, TSI Inc., Minneapolis, MN, USA) to measure the aerosol scattering coefficient

$\sigma_s$ at 550 nm for low humidity ("dry") conditions ($RH_0$) and after exposing the aerosol to humid conditions with relative humidity $RH$, leading to the ratio (Ziemba et al., 2013)

$$f(RH, RH_0) = \frac{\sigma_s(RH)}{\sigma_s(RH_0)}. \tag{25}$$

As discussed by Brock et al. (2016a), the aerosol scattering cross section is roughly proportional to the aerosol volume, and, hence $f(RH)$ is approximate equivalent to the volume growth factor $g_V$ (Eq. 7) when $RH_0 = 0\%$. Based on this approximation

and Eq. 8, Brock et al. (2016a) proposed the parameterization

$$f(RH, 0\%) = 1 + \kappa_{ext} \frac{RH}{1 - RH}. \tag{26}$$

During the CAMP$^2$Ex and ACTIVATE missions, the scattering coefficients are reported at dry conditions with $RH \approx 20\%$ and humid conditions with $RH \approx 80\%$ when both values exceed 5Mm$^{-1}$. Using Eq. 26, this leads to

$$f(80\%, 20\%) \approx \frac{1 + 4\kappa_{ext}}{1 + \kappa_{ext}/4}. \tag{27}$$

For an observed $f(80\%, 20\%)$, $\kappa_{ext}$ can be estimated using Eq. 27. As discussed by Brock et al. (2016a), the parameters $\kappa$ (Eq. 8) and $\kappa_{ext}$ are not the same, but are related and vary approximately proportionally. Their ratio depends on several factors,



including particle size and $\kappa$ itself. Based on the slope of linear fit through observed corresponding $\kappa_{ext}$ and $\kappa$ values reported by Brock et al. (2016a), we use the approximation

$$\kappa \approx \frac{\kappa_{ext}}{0.56}.$$ (28)

Furthermore, based on the values of $\kappa_{ext}$ versus $\kappa$ plotted by Brock et al. (2016a, their figure A3), we estimate a 40% uncertainty of this conversion factor. Subsequently, from the independent DLH measurements of ambient $RH$, the growth factor and thus water volume fraction $f_w$ at ambient condition at the location and time of the in situ observation can be derived using Eqs. 27, 28, 8 and 12.

Probably the greatest source of uncertainty of this approach to derive $f_w$ is the conversion factor between $\kappa_{ext}$ and $\kappa$ (Eq.
28). A 40% under- or overestimation of $\kappa$ leads to a under- or overestimation of $f_w$ that is generally below 0.13. As discussed by Shingler et al. (2016b) and also shown in section 4, $f(80\%, 20\%)$ may sometimes be observed to be below unity. Such values below unity are also commonly observed and generally attributed to measurement uncertainties in the case of aerosol with low hygroscopicity modes (e.g., Holmgren et al., 2014; Kim et al., 2020). We refer to Shingler et al. (2016b) for a discussion on possible other causes. Here, we set $\kappa$ to zero for cases with observed $f(80\%, 20\%) < 1$.

Dry optical size distribution data were observed in situ by the laser aerosol spectrometer (LAS, TSI model 3340, Hilario et al., 2021; Moore et al., 2021). Sizing is corrected during post-flight processing using monodisperse ammonium sulfate aerosol so that derived size distributions are referenced to a real refractive index of 1.53. From these, the integrated particle number concentration, total volume and total surface areas for particle diameters between 100 and 1000 nm are derived. In turn, the effective radii are calculated from the total volume and surface area via Eq. 2. Number concentrations are reported at
a standard pressure and temperature and scaled to the ambient conditions.

Sulfate mass fraction is used as a proxy for soluble fine-mode aerosol mass fraction. Sulfate aerosol mass fractions are derived from observations of the High Resolution Time-of-Flight Aerosol Mass Spectrometer (HR-ToF-AMS, Aerodyne Research, Canagaratna et al., 2007). The inlet maximum diameter cutoff for these observations is effectively sub-micron. The AMS uses thermal vaporization of the particles at $600^{o}C$, electron impaction ionization and provides ensemble-averaged mass
concentrations of non-refractory chemical components. Organic and inorganic components of the aerosols, such as sulfates, are identified through characteristic ion fragments. Sulfate mass fraction $f_{m,sul}$ is calculated as the ratio of total sulfate mass collected over a selected flight segment and the total mass of ammonium, nitrate, sulfate, chloride and organic species of that flight segment (cf. Quinn et al., 2005).

## 4 Results

Here we compare the remote sensing results to corresponding in situ observations. However, in situ and remote sensing results are generally not exactly colocated in space and time, especially when using a single aircraft as was the case during CAMP$^2$Ex. Therefor, we opt for comparisons of campaign-wide and daily statistics of remote sensing and in situ measurements on the premise that statistics of the observations over a similar time range and region should be consistent. For this, parts of flights



were selected during which sufficient remote sensing and in situ data were collected within a $5° × 5°$ region. The dates and
time ranges of the included remote sensing and in situ data are listed in Table S2 and S3, respectively. In situ observations were
collected within the time ranges of the RSP observations or at most 100 minutes before and after these times. Furthermore, only
in situ measurements taken at altitudes below 2 km are used, since most aerosol are expected to be confined to the boundary
layer. Statistics of all data are given in Supplement Section 1.

## 4.1 Retrieval of volume water fraction

Figure 3 show histograms of in situ observations of $f(80\%, 20\%)$ and derived $\kappa_{ext}$ for ACTIVATE winter and summer and
CAMP²Ex, respectively. Values of $f(80\%, 20\%)$ range from about 0.5 to 2.5, with modes near 1.1–1.3. The lowest values,
smallest mean and narrowest distribution are seen for the winter deployment of ACTIVATE. The summer deployment of
ACTIVATE shows a slightly higher mode and broader distribution. During CAMP²Ex, the largest mode is observed and in-
terestingly a second maximum around $f(80\%, 20\%) = 1$ is apparent. Note that the instrumentation used during ACTIVATE
and CAMP²Ex is not able to resolve multi-modal hygrospopicity within a single measurement, while such multi-modal hy-
grospopicity distributions are frequently observed in a range of conditions using other instruments (Heintzenberg et al., 2001;
Swietlicki et al., 2008; Wex et al., 2010; Holmgren et al., 2014; Kim et al., 2020). The continuous $f(80\%, 20\%)$ distributions
observed during ACTIVATE suggest that the observed aerosols are variable mixtures and not representative of a single source
or aerosol type, while the resolved low and high modes observed during CAMP²Ex may be indicative of separately sampling
biomass burning emissions from Borneo and transported Asian emissions, respectively. The many values of $f(80\%, 20\%)$ near
or below unity leads to a strong peak in the distributions of $\kappa_{ext}$ at zero. A second peak is seen near $\kappa_{ext}$ values of 0.05–0.08.
Figure 4 shows that the relative humidity distributions during ACTIVATE displays considerable ranges with multiple peaks,
while the CAMP²Ex relative humidity is generally near 80% representing the humid environment around the Philippines during
late summer.

Figure 5 shows histograms of RSP-retrieved fine-mode refractive indices and water volume fractions derived as discussed
in section 2. Only RSP retrievals with a fine-mode optical depth greater than 0.05 are included. In addition, the water volume
fractions derived from in situ measurements as discussed in section 3.3 are shown. For the winter deployment of ACTIVATE,
the retrieved refractive indices shows two distinct peaks. One peak is near 1.54, corresponding to our assumed refractive index
of dry aerosol (see section 2.3). Another peak is near 1.48, which corresponds to the a priori value of the refractive index in the
optimal estimation scheme of the retrieval algorithm. The peak near the a priori value is also seen during in the results of the
summer ACTIVATE data, albeit to a lesser extend, while it is not apparent for the CAMP²Ex data. For the data of the summer
deployment of ACTIVATE, a broader peak between about 1.50–1.56 is seen, while for CAMP²Ex a broad peak occurs around
1.45 in addition to a weak peak at 1.54.

The prevalence of retrievals around the a priori value of the refractive index indicates that during the ACTIVATE campaign
the information content of the RSP measurements with respect to refractive index was limited for a substantial number of
cases. For most days for both ACTIVATE deployments, there is a prevalence of refractive index retrievals near the a priori
value. Similar statistics are obtained when limiting the data to those with optical depths above 0.2, where information content





may be expected to be higher. We speculate that the reduced information content of some of the ACTIVATE data compared to that of CAMP²Ex may be related to the fact that flight planning during ACTIVATE was constrained by available corridors for civil aviation to fly offshore and a desire in many cases to fly in the direction of the boundary layer wind, while during CAMP²Ex airplane headings were often selected to obtain observations within the principal plane, which maximizes information content for fine mode aerosol retrievals (Fougnie et al., 2020). Furthermore, while cases influenced by cirrus above the aircraft are mostly filtered out by the cost function filter (see section 2, some cases may remain in the dataset possibly causing the prevalence of refractive index retrievals near the a priori value. However, more investigation is needed into why the information content with respect to refractive index may have been lower for part of the retrievals of ACTIVATE.

The water volume fractions derived from the retrieved refractive indices show a larger peak at zero for all three campaigns, corresponding to retrieved refractive indices $\geq 1.54$. For the ACTIVATE campaigns, a peak near $f_w = 0.3$ is seen, which corresponds to the a priori value of the refractive index, as discussed above. The figures also show the water fractions estimated from the in situ observations, as described in section 3.3. Similarly to the RSP retrievals, these also show a large peak at zero, corresponding to $\kappa = 0$. For the ACTIVATE data, the frequency of in situ observed water fraction generally decreases with its value, while the CAMP²Ex results show a broader distribution peaking around $f_w = 0.45$, similarly to the RSP distribution. Using the Kolmogorov-Smirnov test reveals a 89% likelihood that the in situ and RSP water fractions from the CAMP²Ex data are drawn from the same distribution. For the ACTIVATE data this theoretical likelihood calculated with the Kolmogorov-Smirnov test is 3%, presumably because of the prevalence of retrieved refractive index near a priori values.

Next, we investigate the daily mean water volume fraction obtained from the RSP and in situ measurements. Figure 6 shows that the RSP and in situ values compare reasonably, although the spread is greater than the uncertainty in RSP retrievals expected from the assumed $\pm 0.02$ uncertainty in dry refractive index. However, given the estimated uncertainty in in situ-derived water fraction of about 0.13 and the standard deviations of the daily values that are 0.21 and 0.20 for the in situ and RSP-retrieved values, respectively, the comparison may be considered favorable. Mean absolute difference between all RSP and in situ means is 0.09 with a correlation coefficient of 0.68. Daily values for CAMP²Ex show the best agreement between RSP and in situ water fractions with a mean absolute difference 0.080 and a correlation coefficient of 0.82.

## 4.2 Retrieval of soluble aerosol fraction

As discussed in section 2, effective radii and variances are expected to increase with volume water fraction depending on the fraction of soluble aerosol. Examples of retrieved effective radii and variances as a function of derived volume water fraction are shown in Figs. 7 and 8. On 2 September 2020 during ACTIVATE, a clear increase of both effective radius and variance with volume water fraction is seen, although a substantial spread is also observed. On 21 September 2019 during CAMP²Ex, the increase of effective radius and variance with volume water fraction is also apparent but weaker. The spread in the data may be attributable to natural variation of aerosol dry size distributions during the observation period, as well as uncertainties in effective radius retrievals and water fraction estimates. As discussed in section 2.2, an approximately linear relationship between $v_e + 1$ and $r_e$, both relative to their dry values is expected. The reference dry values for each day, here indicated as $v_{e,ref}$ and $r_{e,ref}$, are estimated from the retrieved $v_e$ and $r_e$ values for water fractions $0 < f_w < 0.2$, which are assumed to



represent nearly dry but partly soluble aerosol. To estimate $v_{e,ref}$ and $r_{e,ref}$, we fit modeled $v_e$ and $r_e$ as a function of $f_w$ for a default soluble fraction of $f_{sol} = 0.3$ (via Eqs. 21 and 20) to these daily cases with $0 < f_w < 0.2$. The modeled $v_{e,dry}$ and $r_{e,dry}$ values associated with the lowest root-mean-squared difference between model and data are used as $v_{e,ref}$ and $r_{e,ref}$,

respectively, for that day. Rather than taking averages of the $v_{e,ref}$ and $r_{e,ref}$ observations at $0 < f_w < 0.2$, this approach allows to approximately account for the slight variation of $v_e$ and $r_e$ with $f_w$. Using a model for other values of $f_{sol}$ for this estimation of $v_{e,ref}$ and $r_{e,ref}$ yields similar overall results. The bottom panels of Figs. 7 and 8 show the retrieved $v_e + 1$ and $r_e$ values relative to $v_{e,ref} + 1$ and $r_{e,ref}$ for all retrievals with $f_w > 0$. The expected approximately linear relationship is observed for both cases, with a steeper slope for the ACTIVATE case (Fig. 7) compared to the CAMP$^2$Ex case (Fig. 8) . Subsequently,

we find the lowest root-mean-square difference between the observations of $(v_e + 1)/(v_{e,ref} + 1)$ versus $r_e/r_{e,ref}$ and values in a pre-calculated look-up-table with values of $v_e + 1$ and $r_e$ relative to their dry values modeled according to section 2.2 for various $f_{sol}$. A minimum of $f_{sol}$ of 0.05 is assumed, as for lower values unrealistically large growth factors $g_{V,sol}$ need to be assumed to obtain considerable $f_w$. The resulting best fits for ACTIVATE and CAMP$^2$Ex cases are for $f_{sol}$ values of 0.23 and 0.62, respectively, and the corresponding modeled relationships are shown in the bottom panels of Figs. 7 and 8. For reference

and a consistency check, the modeled $r_e$ and $v_e$ as a function of $f_w$ corresponding to the inferred $f_{sol}$, as given by Eq. 20 and Eq. 21, respectively are also shown in Figs. 7 and 8, where $r_{e,dry} = r_{e,ref}$ and $v_{e,dry} = v_{e,ref}$. Note that the retrieval of $f_{sol}$ is mostly constrained by the retrieved $r_e$ and $v_e$ values and only relies on the retrievals of $f_w$ to select cases with low $f_w$ in order to derive the $r_{e,ref}$ and $v_{e,ref}$.

This approach is applied to the daily data of ACTIVATE and CAMP$^2$Ex. Only days for which at least two data points with

$0 < f_w < 0.2$ and at least two with $f_w > 0.2$ are available are selected, which excludes six days from CAMP$^2$Ex. In addition, as a proxy for soluble aerosol contribution to the fine mode, we derive the daily average mass fraction of sulfates from the AMS data as described in section 3.3. Note that the sulfate mass fraction can be interpreted as the fraction of all sulfate that is externally and internally mixed in all aerosols analyzed by the AMS during the selected time period, while the RSP soluble fraction is an estimate of volume fraction of fine mode aerosol that has a $\kappa$ value greater than zero. Hence, they may be expected

to correlate, but are not fundamentally the same.

Figure 9 shows the daily RSP retrievals of soluble aerosol fraction versus in situ-observed sulfate mass fractions for the three campaigns. The soluble fraction is generally low, especially during ACTIVATE. During the winter deployment of ACTIVATE, all days have $f_{sol} < 0.2$, while values are all below 0.4 during the summer deployment. The most spread of $f_{sol}$ is seen during CAMP$^2$Ex with values up to 0.73. These differences between campaigns are qualitatively consistent with the differences in

$f(80\%, 20\%)$ and $\kappa_{ext}$ as seen in Fig. 3.

Turning our attention to the in situ observations of mass fraction of sulfates shown in Fig. 9, it can be seen that sulfate mass fractions between 0.15 and 0.4 are observed during ACTIVATE, while a larger range is seen during CAMP$^2$Ex. Most of the remaining aerosol mass generally consists of organics (not shown). The soluble aerosol fraction retrieved from RSP reasonably correlates with the sulfate mass fraction with a Pearson correlation coefficient of 0.79, although the number of points is low

and the correlation is mostly driven by CAMP$^2$Ex results. A linear fit of the form $f_{sol} = 1.09 \times f_{m,sul} - 0.12$ is also shown in Fig. 9. While the fitted slope is close to unity, we may not expect the RSP-retrieved soluble fraction to perfectly correlate with





the sulfate mass fraction with a 1:1 dependence as a) other species, including organics, may contribute to the total soluble mass fraction, and b) the RSP retrievals assume an external mixture of soluble and insoluble particles while the AMS vaporizes all particles and observes the sulfate mass fraction irregardless of original mixing state. We may, however, expect the intercept of

the linear fit to be near zero as a population with negligible contribution of soluble particles would also be expected to contain negligible sulfate mass. The fact that the intercept is -0.12 indicates that the RSP soluble fraction is underestimated when the real soluble fraction is low. Further discussion is provided in section 5.

### 4.3   Retrieval of dry aerosol size distribution

For known volume water fraction and soluble fraction, the relative increase of effective radius and $v_e + 1$ is given by Eqs. 20

and 21, respectively. Using the volume water fraction retrieved for each observations along with the daily values of soluble fraction, we calculate the inferred dry effective radius and variance for all data from the three campaigns, of which histograms are shown in Fig. 10. Note that for the six days from CAMP²Ex for which retrievals of soluble fraction is not possible we assume $f_{sol} = 0.3$. No noticeable difference in the histogram is seen if a different value is assumed is made.

Largest values of both effective radius and variance is seen during the summer deployment of ACTIVATE and smallest

values during its winter deployment. Differences between ambient and dry values of effective radius and variance are smallest for the winter deployment of ACTIVATE, namely respectively 0.013 $\mu$m and 0.022 on average, consistent with the low volume water and soluble fractions seen in that deployment. Average differences between ambient and dry effective radii and variances for the winter ACTIVATE deployment are 0.020 $\mu$m and 0.028, respectively, while largest differences of respectively 0.034 $\mu$m and 0.042 on average are seen for the CAMP²Ex data, consistent with broader distributions of volume water fraction and

soluble fraction.

Dry effective radii are estimated from the in situ-measured total volume and surface areas as described in section 3.3. These reported volume and surface areas pertain to the particles with dry radii between 0.05 and 0.5 $\mu$m. Hence, the RSP dry effective radii are scaled to include only particles with radii between 0.05 and 0.5 $\mu$m in order to directly compare them to LAS observations. Figure 11 show histograms for the three campaigns of RSP and LAS-derived dry effective radii. The ranges of in

situ-observed effective radii and those retrieved by RSP agree well, although mean values observed in situ are generally smaller (see also Table S1). Largest differences are seen for the summer deployment of ACTIVATE, while best agreement is seen for the CAMP²Ex campaign. Daily means values of in situ-observed and RSP-retrieved dry effective radii for radii between 0.05 and 0.5 $\mu$m are compared in Fig. 12. The mean absolute difference between daily averages is 0.024 $\mu$m. However, the spread is generally large and the correlation is rather poor. The best comparison is for CAMP2Ex with values agreeing within 0.019

$\mu$m on average. Note that the daily standard deviations of effective radius is 0.02 or larger for most days (see Table S1 in the Supplement). Further discussion on the comparisons of dry effective radii and possible biases in RSP and in situ observations is provided in section 5.





### 4.4 Retrieval of aerosol number concentration in given dry size ranges

Column-averaged fine mode aerosol number concentrations $N_{f,tot}$ are estimated by the RSP using Eq. 24. Knowledge of the
dry particle size distribution allows estimating the number concentrations within a given dry particle size range. This allows
remotely sensed number concentrations to be adjusted to take into account in situ probe size limits for better comparison with
in situ observations. Furthermore, as small aerosol are less effective CCN, the variation in $N_{f,tot}$ larger than a given radius limit
$r_{lim}$ can be used as proxy for variation in CCN concentrations (Dusek et al., 2006; Hasekamp et al., 2019). However, using
ambient size distributions in lieu of dry size distribution retrievals in this approach may lead to a bias in estimated variation in
CCN concentrations related to water uptake Hasekamp et al. (2019).

     Figure 13 shows the mean and standard deviation of the fraction of fine mode particles larger than a dry minimum radius $r_{lim}$
for the three campaigns. Fractions are close to unity for $r_{lim}$ up to about 0.04 $\mu$m, but then steadily decrease with increasing
$r_{lim}$ up to about 0.2 $\mu$m. Fractions at a given $r_{lim}$ are largest for the ACTIVATE summer campaign because of the relatively
large $r_{e,dry}$ values at that campaign. The $r_{lim}$ values at which 50% of the particles are larger than $r_{lim}$ are 0.095, 0.010 and
0.089 $\mu$m for the ACTIVATE winter, summer and CAMP$^2$Ex campaigns, respectively.

     The LAS in situ instrument provides number concentrations for particles with dry radii between 0.05 and 0.5 $\mu$m. Daily
geometric mean number concentrations derived from the in situ observations and RSP are compared in Fig. 14. Here, RSP
$N_{f,tot}$ values are scaled using $r_{lim} = 0.05$ $\mu$m, although this scaling has limited effect as can be concluded from Fig. 13.
Generally RSP and in situ daily means agree within a factor 2 and correlate with a correlation coefficient of 0.86. The mean
absolute difference for all days is 138 cm$^{-3}$. Note that the geometric standard deviation factor often exceeds 2 for the in
situ observations of number concentrations, while it is generally around 1.5 for the RSP retrievals (see Table S1). Similar
comparisons between RSP and LAS number concentrations have been shown by Schlosser et al. (2022).

### 5 Discussion on assumptions and uncertainties

The presented methods to infer volume water fraction and daily soluble fraction are largely independent from each other. The
derivation of dry effective radius from its ambient counterpart is mostly relying on the retrieved volume water fraction, while
the derivation of dry effective variance also substantially depends on estimated soluble fraction.

     The volume fraction of water in the observed aerosol is inferred from the retrieved ambient refractive index using the
assumptions that the volume mixing rule applies to the refractive index of internally and externally mixed aerosol and that
the dry refractive index is 1.54. While both these assumptions lead to uncertainties, we estimate that these uncertainties in the
retrieved volume water fraction are generally below about 0.2 and decrease with increasing volume water fraction. Furthermore,
while substantially deviations from dry refractive indices of 1.54 occur in individual in situ observations, the standard deviations
for data collected over regions or periods are generally small (e.g., Aldhaif et al., 2018), suggesting the accuracy of retrieved
volume water fraction generally improves upon averaging. However, for regions and periods with aerosol mixtures that have
average dry refractive index systematically smaller than 1.5, the presented method will yield biased results. Furthermore, in



case of pure salt aerosol the use of the volume mixing rule may lead to an overestimation of volume water fraction if the real water fraction is below about 50% (Schuster et al., 2009, see also Supplement section 2).

We further assume that the aerosol consists of externally mixed soluble ($\kappa > 0$) and insoluble ($\kappa = 0$) components with equal dry size distributions. While this model explains the co-variation of retrieved fine-mode effective radius and variance reasonably well and is based on previous in situ observations under many conditions with instruments that can resolve multi-

modal hygroscopicity, it is considered a practical, over-simplified model. In reality, insoluble and soluble components have distributions of (effective) $\kappa$ values around zero and non-zero values, respectively (Heintzenberg et al., 2001; Holmgren et al., 2014; Kim et al., 2020). Furthermore, the approach to estimate the soluble aerosol fraction uses daily data and makes the assumption that the soluble aerosol fraction is fixed during the observation period. While the retrieved daily soluble fractions reasonably correlate with in situ estimated sulfate mass fraction, the method yields many soluble fractions below 0.2, while the

sulfate mass fractions are generally greater than 0.2 (See Fig. 9). We speculate that the over-simplified aerosol model used here may be biasing results low if the soluble component in reality has a broad distribution of $\kappa$ values further broadening the size distributions. Furthermore, Holmgren et al. (2014) show hygrospicity generally increases with size, which is inconsistent with our assumption of equal size distributions for the soluble and insoluble components and may promote increased broadening of the size distribution upon humidification, in turn resulting in a low bias in estimate soluble fraction. Future studies based on

simulated measurements are needed to further study these effects.

We also assume that all soluble particles are hydrated at the ambient conditions, i.e. that the RH is above the efflorescence and/or deliquescence points for all soluble particles. If in reality a substantial fraction of the soluble aerosol is not deliquesced, this may bias the estimated soluble fraction low. However, there is no correlation apparent between daily-average RH and inferred soluble fraction (See supplement Tables S1 and S3). Furthermore, the $f(80\%, 20\%)$ observations confirm the generally

low hygroscopicity, although these are not affected by the ambient RH.

Another implicit assumption made in the method is that all observed aerosol is exposed to a similar relative humidity. While most aerosol is confined to the boundary layer which is generally well-mixed, the relative humidity may be expected to increase with decreasing temperature and thus with altitude. A substantial variation in relative humidity in the aerosol-loaded column may contribute to a broadening of the column-averaged size distributions and possibly a bias in inferred soluble

fraction. We perform a simple test whether the RSP-retrieved soluble fractions and ambient size distributions are consistent with the relative humidity observed in situ below 2 km altitude. For this, we first assume a $\kappa_{sol}$ value of 0.5 for the soluble fraction, which is representative of ammonium sulfate (Petters and Kreidenweis, 2007), and estimate an aerosol average $\kappa$ from $\kappa = \kappa_{sol} \times f_{sol}$. Subsequently, the relative humidity can be estimated for each observation using Eq. 26 and the growth factor derived from the ambient $r_{eff}$ relative to the estimated $r_{eff,dry}$ (see section 4.2). This yields daily mean relative humidity

values that are within 3.5%, 3.2% and -4.8% (absolute values) of the daily mean in situ-observed values for the ACTIVATE winter and summer deployments and the CAMP²Ex campaign, respectively. This generally good agreement suggests an overall consistency between the soluble fraction and ambient size distributions from RSP and the relative humidity observed in situ below 2 km altitude.





Using the volume water fraction and soluble fractions the dry effective radii and variances were estimated from the ambient

RSP retrievals. While the total ranges of RSP-derived and in situ-measured dry effective radius are consistent, the in situ values are substantially smaller on average and RSP and in situ daily averages correlate rather poorly, especially for the ACTIVATE deployments. This may suggest a substantial low bias in the RSP-retrieved volume water fraction, although this is unlikely considering its generally favorable comparison to water fractions derived from in situ observations. Furthermore, a positive bias in the retrievals of ambient effective radius may possibly be the culprit. Alternatively, the LAS observations of total

volume and surface area may be biased low or high, respectively. There are indications that relatively large aerosol particles are 'lost' in the tubing system of the aerosol observation package, which would be consistent with a low bias in effective radius. Further analysis of these and future LAS data is needed to evaluate this potential issue. Furthermore, further evaluation of the dry effective radius derived from RSP data with the presented method is needed using other past or future campaign data.

## 6 Conclusions

We present a framework to infer volume water fraction, soluble fraction and dry size distributions from multi-angle, multi-spectral polarimetry retrievals of column-averaged fine mode ambient aerosol properties. Volume water fraction is inferred from the ambient refractive index, while average soluble fraction is mostly derived from size distributions within a region or time period. Both volume water fraction and soluble fractions are used to infer dry size distributions from their ambient counterparts. The approach is applied to observations of the RSP during the ACTIVATE and CAMP$^2$Ex field campaigns and

compared to in situ observations obtained below 2 km.

The daily-averaged volume water fractions from RSP and in situ observations show good correlation and have a mean absolute difference of 0.09. Daily estimates of soluble fraction correlates reasonably well with in situ observed sulfate mass fraction, although the soluble fraction appear to be biased low. RSP-derived dry effective radius shows a similar range as in situ observed values. However, during the ACTIVATE deployments, in situ derived effective radius is generally smaller

by about 0.02-0.05 $\mu$m. Possible causes of this inconsistency may be related to RSP or in situ observations. RSP-retrieved number concentrations generally agree well with the in situ observations. Best agreements overall are seen for the CAMP$^2$Ex campaign. The ACTIVATE RSP data appear to be hampered by lower information content possibly caused by non-favorable flight direction or cirrus above the aircraft leading to retrievals of refractive index to be frequently strongly weighted towards the a priori value. We note that the RSP and in situ observations are not strictly co-located in space and time but are compared

on the premise that mean values and standard deviations of the observations over a similar time range and region should be consistent.

Both RSP and in situ observations indicate the dominance of aerosol with low hygroscopicity during the ACTIVATE and CAMP$^2$Ex campaigns. Furthermore, RSP indicates a high degree of external mixing that is not resolved by the in situ observations. This is consistent with high degree of external mixing of modes with low and high hygroscopicity shown by Swietlicki

et al. (2008) and Holmgren et al. (2014) and others for various aerosol sources. Furthermore, Holmgren et al. (2014) showed that the contribution of low hygroscopicity particles is seasonally varying, peaking in winter, which may be consistent with the



relatively low water and soluble fractions found in the ACTIVATE winter deployment. The hydrophobic or insoluble particles may consist of long-lived and externally mixed primary (Petters and Kreidenweis, 2007) or secondary (Pun et al., 2002) organic aerosol or 'tar balls' that have their origin in biomass burning (Pósfai et al., 2004; Yuan et al., 2021). Such particles may

originate from fires that were prevalent in the ACTIVATE (Corral et al., 2021) and CAMP2Ex (Hilario et al., 2021; Reid et al., 2022) areas.

The remote sensing of fine mode water volume fraction may be used for evaluation of water uptake in atmospheric models. We also demonstrate how the derived dry size distributions can be used to derive the fraction of fine mode particles within a give dry particle size range. This is relevant for relating retrieved total fine mode particles to CCN concentrations, as it has

been shown that variations in CCN concentrations may better correlate with concentrations of aerosol above a certain size rather than the concentrations of all aerosol (Dusek et al., 2006; Hasekamp et al., 2019). Knowledge on the hygrospopicity of the particles and the soluble fraction of externally mixed aerosol may also be important for estimating CCN from total aerosol concentrations (Wex et al., 2010), although insoluble particles may still act as effective CCN (Kumar et al., 2009).

The presented approach may be applied to any multi-angular, multi-spectral polarimeter observations over land and ocean

that are sufficiently accurate enough to infer fine-mode refractive index, effective radius and variance (Mishchenko et al., 2004; Hasekamp and Landgraf, 2007). The HARP-2 and SPEXone (Hasekamp et al., 2019) instruments on NASA's PACE satellite mission (Werdell et al., 2019) to be launched in 2024 are expected to provide such observations at a spatial resolution of about $5 \times 5$ km$^2$. The derivation of water volume fraction may be directly applied to pixel-level satellite retrievals of refractive index, while regional data may be used to estimate soluble fraction. Furthermore, pixel-level dry size distributions may be computed

from the retrieved ambient values according to the availability of successful water volume and soluble fraction retrievals.

In conclusion, we have shown that aerosol water volume fraction may be derived from the retrieved ambient refractive index with an uncertainty estimated to be better than 0.2 and which decreases further with increasing volume water fraction. For regional data, the retrieved particles sizes increase with water volume fraction as expected, further bolstering confidence in the water fraction retrievals. Furthermore, the observed particle size distribution width generally also increases with water fraction,

which, as we show, points to the presence of external mixtures of soluble and insoluble aerosol particles. To our knowledge, these results represent the first airborne remote sensing observations of aerosol water volume fraction that are evaluated with corresponding in situ measurements. Application of this approach to global satellite observations may be useful to better constrain aerosol water uptake represented in models and to improve estimates of CCN concentrations from remote sensing.

*Data availability.* The datasets analyzed for this study can be found in the NASA LaRC Airborne Science Data for Atmospheric Composition

(https://www-air.larc.nasa.gov/). Digital Object Identifier (DOI) numbers for the datasets are 10.5067/SUBORBITAL/ACTIVATE/DATA001 and 10.5067/Suborbital/CAMP2EX2018/DATA001.



*Author contributions.* BD, OH and GS developed the method. SS, BC, MS, and LZ provided and curated data. BD analyzed the data. BD prepared the manuscript with contributions from all co-authors.

*Competing interests.* No competing interests are present

*Acknowledgements.* Funding support for this work was provided 1) by the Dutch Research Council (NWO) and Netherlands Space Office (NSO) under grant. no. ALWGO.2019.006 (PACE-ACI: Using the PACE mission for improved quantification of aerosol-cloud interactions); 2) through NASA grant no. 80NSSC18K0150 in support of the CAMP$^2$Ex campaign and 3) through the ACTIVATE Earth Venture Suborbital-3 (EVS-3) investigation, which is funded by NASA's Earth Science Division and managed through the Earth System Science Pathfinder Program Office. We thank all involved with the CAMP$^2$Ex and ACTIVATE campaigns.





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




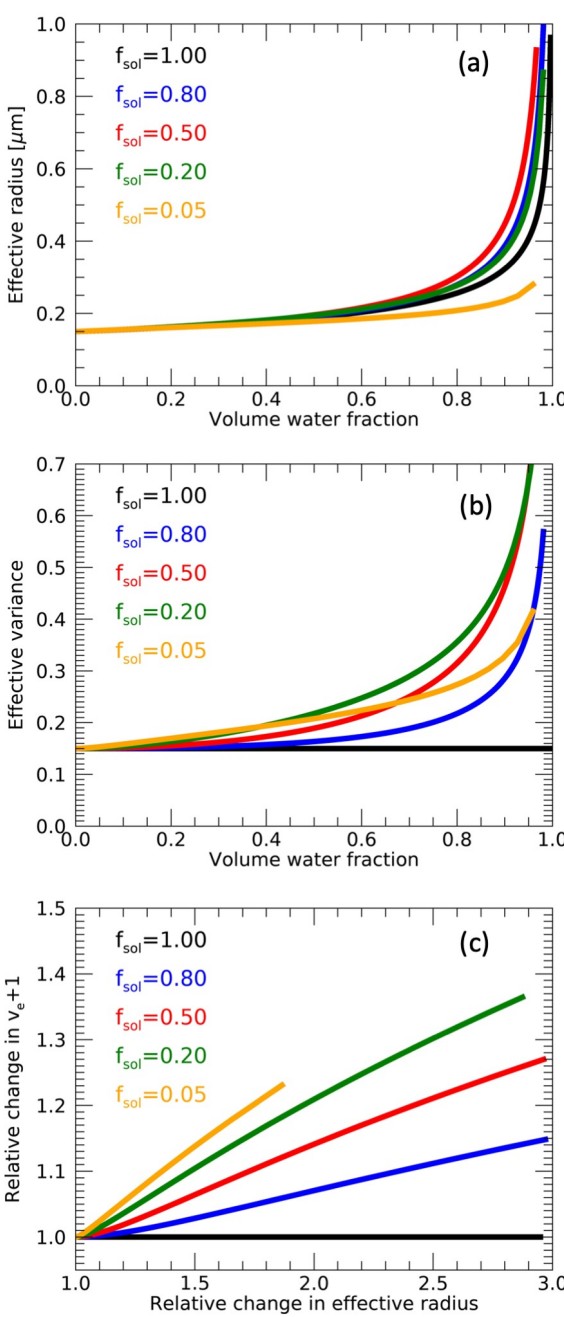

**Figure 1.** Modeled effective radius (a) and effective variance (b) of an aerosol size distribution as a function of water volume fraction for a fully soluble aerosol (black, $f_{sol} = 1$) and for external mixtures of insoluble and soluble aerosol with different soluble fractions $f_{sol}$ represented by different colors. Here, $r_{e,dry} = 0.15\,\mu$m and $v_{e,dry} = 0.15$. Panel (c) shows the relative change in $v_e + 1$ versus relative change in effective radius for different values of $f_{sol}$.





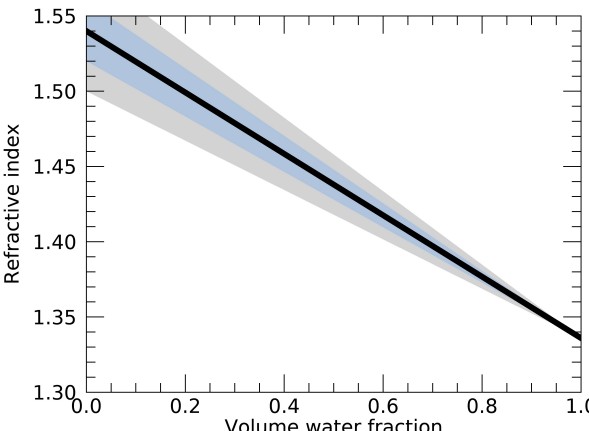

**Figure 2.** Modeled aerosol refractive index versus volume water fraction when assuming a volume-weighted average of the refractive index of water and a dry aerosol refractive index of 1.54 (solid line). The blue shaded area indicates the range between results when dry refractive index is 1.52 and 1.56, respectively, while the grey shaded area indicates the range between results when dry refractive index is 1.50 and 1.58, respectively.







**Figure 3.** Histograms of the in situ observations of $f(80\%, 20\%)$ (left) and $\kappa_{ext}$ (right) obtained during the ACTIVATE winter (top) and summer (middle) deployments and CAMP$^2$Ex (bottom). The dashed line in the left plots indicates $f(80\%, 20\%) = 1$.




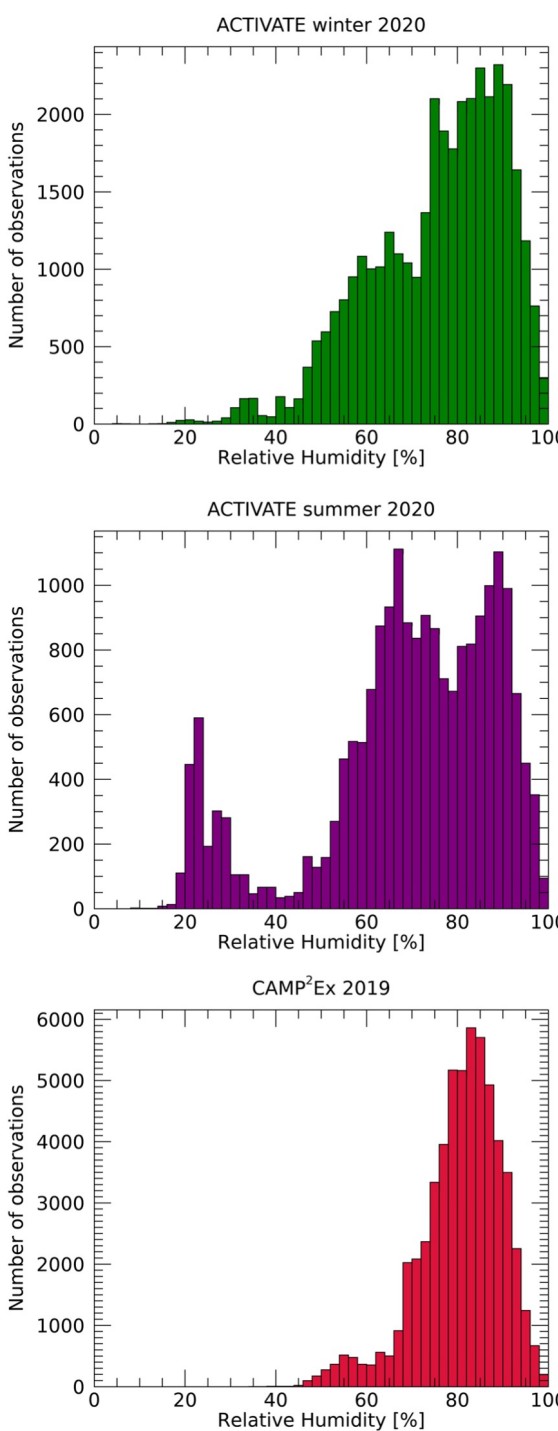

**Figure 4.** Histograms of the in situ observations of relative humidity obtained during the ACTIVATE winter (top) and summer (middle) deployments and CAMP²Ex (bottom).



**Figure 5.** Similar to Fig. 3, but showing the RSP-retrieved refractive index (left) and derived volume water fraction (right). Grey bars in the right panels show the volume water fraction derived from the in situ observations. Volume water fraction histograms are normalized to their maximum value. The dashed lines in the left panels indicate the assumed $n_{dry} = 1.54$.





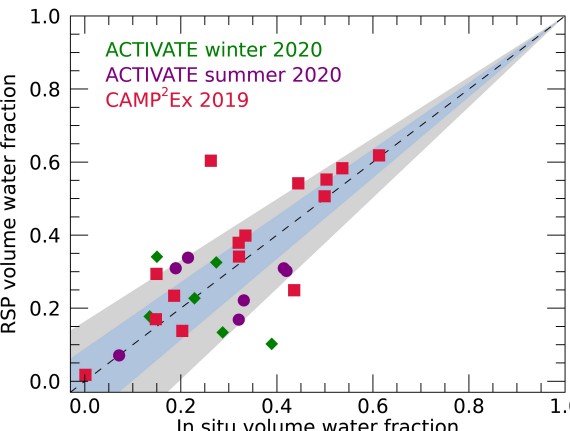

**Figure 6.** Daily mean volume water fractions retrieved from RSP versus those derived from the in situ data. Data from the ACTIVATE Winter, Summer and CAMP[2]Ex campaigns are indicated by green diamonds, purple circles and red squares, respectively. The blue and grey shaded regions show the expected accuracy of the RSP retrievals based on an uncertainty in $n_{dry}$ of $\pm 0.02$ and $\pm 0.04$, respectively, as also indicated in Fig. 2.





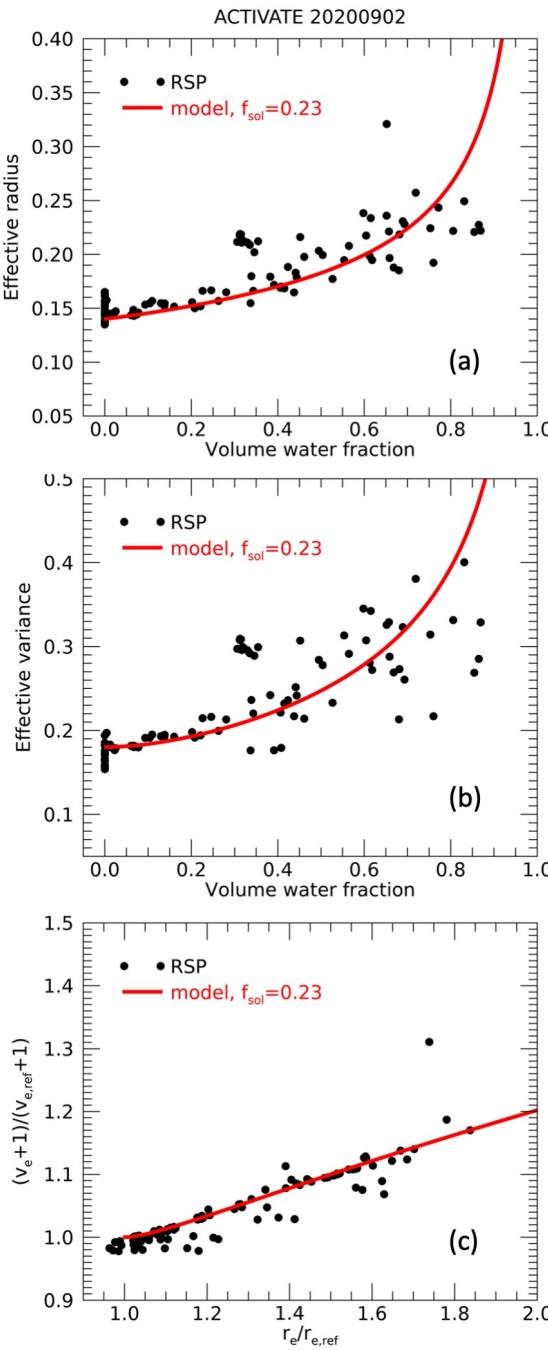

**Figure 7.** Examples of effective radius (top) and variance (middle) as a function of volume water fraction for an ACTIVATE case with relatively low retrieved soluble aerosol fraction. The bottom figure shows the ratio $(v_e + 1)/(v_{e,ref} + 1)$ versus the ratio $r_e/r_{e,ref}$ (see text) for cases with $f_w > 0$. Red lines in all panels show the model values that correspond to the soluble fraction that leads to the best fit to the $(v_e + 1)/(v_{e,ref} + 1)$ versus $r_e/r_{e,ref}$ data points.




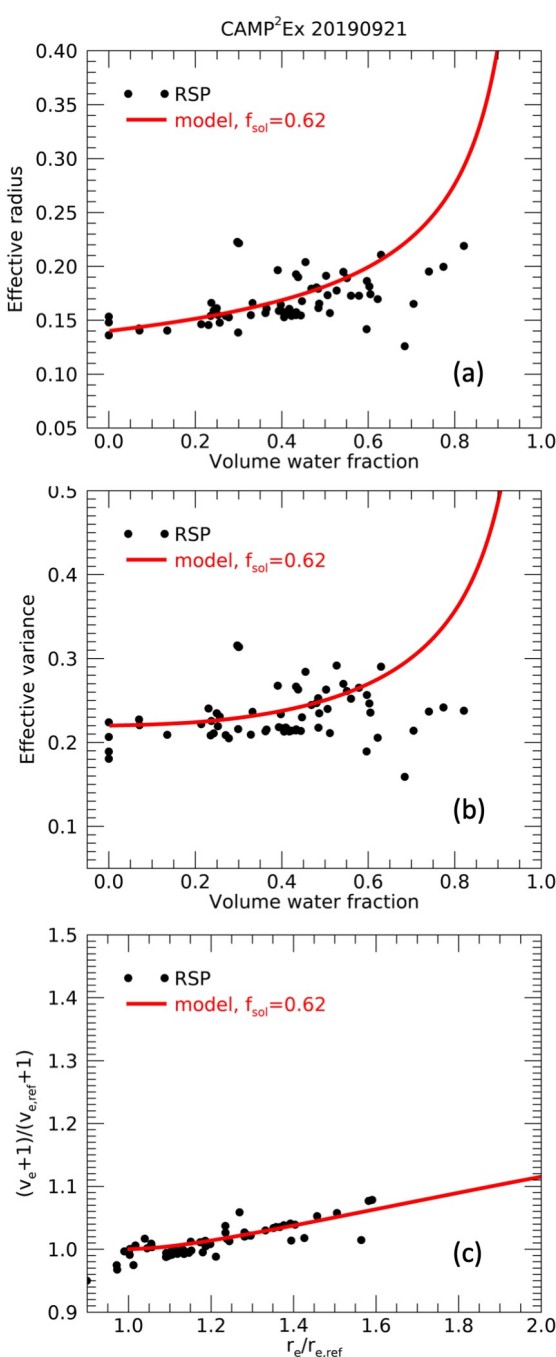

**Figure 8.** Same as 7 but for a case with relatively high retrieved soluble aerosol fraction during CAMP²Ex.





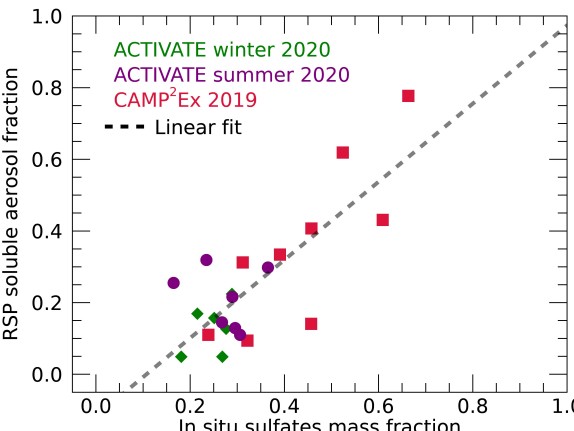

**Figure 9.** Daily RSP retrievals of the soluble aerosol fraction $f_{sol}$ versus daily in situ measured sulfate aerosol mass fractions. The dashed line shows a linear least-squares fit ($f_{sol} = 1.09 \times f_{m,sul} - 0.12$) through all points. Data from the ACTIVATE Winter, Summer and CAMP$^2$Ex campaigns are indicated by green diamonds, purple circles and pink squares, respectively.





**Figure 10.** Histograms of RSP-retrieved fine-mode effective radii (left) and effective variance (right) obtained during the ACTIVATE winter (top) and summer (middle) deployments and CAMP$^2$Ex (bottom). Blue bars represent ambient values, while colored bars indicate the corresponding dry values derived as described in the text.





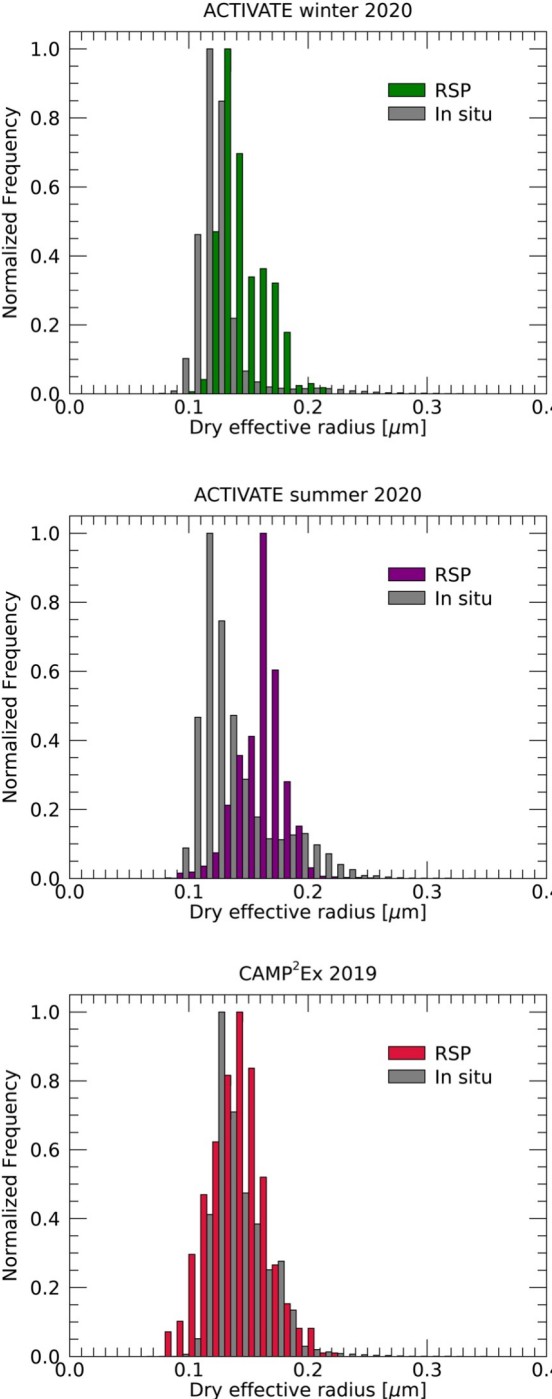

**Figure 11.** Histograms of dry fine-mode effective radii retrieved by RSP (colored bars) and those derived from in situ data (grey) for the ACTIVATE winter (top) and summer (middle) deployments and CAMP²Ex (bottom). Effective radii radii are computed including only particles with radii between 0.05 and 0.5 $\mu$m. The histograms are normalized to their maximum value.



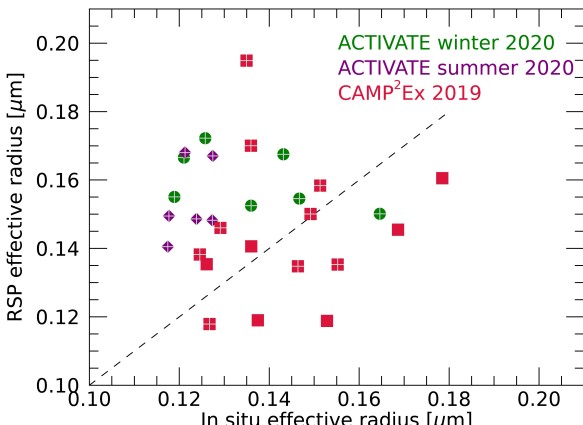

**Figure 12.** Daily means of dry fine-mode effective radii retrieved by RSP and those derived from in situ data for the three campaigns. Effective radii radii are computed including only particles with radii between 0.05 and 0.5 $\mu$m. Data from the ACTIVATE Winter, Summer and CAMP[2]Ex campaigns are indicated by green diamonds, purple circles and red squares, respectively. Days with a successful retrieval of soluble aerosol fraction are marked by a plus.





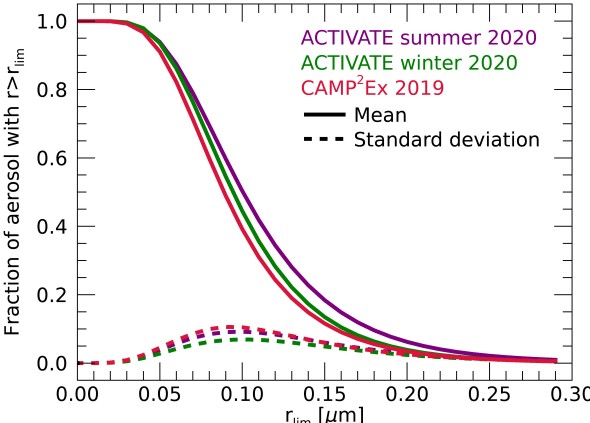

**Figure 13.** The mean (solid lines) and standard deviation (dashed lines) of the fraction of fine mode particles larger than a dry minimum radius $r_{lim}$ for the three campaigns.





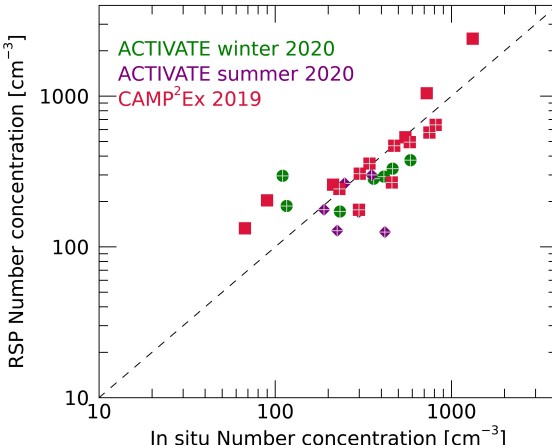

**Figure 14.** Daily means of number concentrations retrieved by RSP and those derived from in situ data for the three campaigns. Data from the ACTIVATE Winter, Summer and CAMP$^2$Ex campaigns are indicated by green diamonds, purple circles and red squares, respectively. Days with a successful retrieval of soluble aerosol fraction are marked by a plus.