# Peer review of "Remote sensing of aerosol water fraction, dry size distribution and soluble fraction using multi-angle, multi-spectral polarimetry"

_EGUsphere, 2022_

## Author Response (AR1)

**Reviewer 1**

We thank the reviewer for the for the positive feedback and helpful comments. Our responses are indicated in red below.

1. Line 150: The authors assumed the size distributions of the insoluble component and the dry soluble component of the mixture are the same. Is there any back up reference?

Little information is available. We added a discussion on this assumption and a sensitivity study at line 183 and further. In the discussion we include the sentence:

"Some observations, such as from Holmgren et al. (2014) and Kim et al. (2020), suggest that hygroscopicity increases with particle size."

Please also see our reply to comment 5.

2. Line 254:Why only data over ocean are used in this manuscript? Please specify.

To specify we added the following to the text at line 275:

"The majority of the data during these campaigns were collected over ocean. Furthermore, the RSP aerosol retrieval algorithm (section 3.2) is currently limited to ocean surfaces. Hence, only data over ocean are used here."

3. Section 4.1: The Figure 5 showed that the water volume fractions derived from the retrieved refractive indices show a larger peak at zero for all three campaigns, corresponding to $\kappa = 0$. Why the retrieved aerosol hygroscopicity can equal to zero? As I understand, even the even the water-insoluble particles will be somewhat hygroscopic rather than absolutely not hygroscopic.

Petters and Kreidenweis (2007) define kappa as equal or greater than 0 (e.g., see their Fig 1). Note that these insoluble particles can be wettable at supersaturation above 1, as discussed by Petters & Kreidenweis: "As kappa approaches zero, the particle becomes nonhygroscopic and the slope approaches that expected for an insoluble but wettable particle as predicted by the Kelvin equation, i.e. –1."

Since we already refer to Petters and Kreidenweis (2007) for our definition of kappa, no changes were made to the manuscript related to this comment.

4. Line 448: Why is there no difference in the retrieved results assuming different f values? This is confused. Because the solube aerosol fraction has lagre effect on aerosol hygroscopicity and thus influence particle size distribution.

The reviewer is correct that soluble fraction has a large effect on the aerosol growth as a function of relative humidity. However, note that we derive dry effective radius and variance from the ambient values using the simulated relationship between r_e and v_e and *water fraction* in combination with the retrieved water fraction. Figure 1 shows that the relationship between r_e and water fraction is rather similar for f_sol values >0.05. The relationship

between v_e and water fraction varies somewhat more with f_sol, but this variation is still limited for f_sol between about 0.1 and 0.7.

To further clarify, we added the following to the text at line 473:

"Since the variation of re,mix and ve,mix with fw vary relatively weakly for $0.1 < fsol < 0.6$ (Fig. 1), no noticeable difference in the histogram is seen if a different value fsol is assumed within this range."

5. Section 5: Same above. the authors assume that the aerosol consists of externally mixed soluble ($\kappa > 0$) and insoluble ($\kappa = 0$) components with equal dry size distributions. This will lead to large uncertainty in retrieved results. I suggest the uncertainty analysis about this assumption are needed.

According to the reviewer's suggestion we added a sensitivity study to the supplement (section S2) and summarize the results in the main text. The following was added to the text starting at line 183 and further:

"To derive Eqs. 18–21, we assumed that the size distributions of the insoluble component and the dry soluble component of the mixture are the same. Some observations, such as from Holmgren et al. (2014) and Kim et al. (2020), suggest that hygroscopicity increases with particle size. To test the impact on assuming the same dry particle distribution of insoluble and soluble aerosols, we simulated re,mix and ve,mix as a function of fsol for a range of fw and for an aerosol with a soluble component with re,dry = 0.15 µm and an insoluble component that is Δre smaller. For both components we assume ve,dry = 0.15. Using these simulations as our dataset, we subsequently infer fsol, re,dry and ve,dry while assuming an aerosol mixture with equal soluble and insoluble components. The differences between the inferred fsol, re,dry and ve,dry and the true values in the simulations yield an estimated sensitivity to Δre. Detailed results are given in the Supplement. For a Δre of 0.03 µm, maximum underestimations of fsol, re,dry and ve,dry are 0.23, 9% and 17%, respectively, occurring at true fsol values of 0.65, 0.28 and 0.16, respectively. These biases scale approximately linearly with Δre. Hence, especially retrieved fsol and ve,dry are quite sensitive to our assumption of the same dry particle distribution of insoluble and soluble aerosols. However, a realistic estimate of the range of Δre is not available."

**Reviewer 2**

We thank the reviewer for the for the positive feedback and helpful comments. Our responses are indicated in red below.

Line 110. How low is the kappa and how small is the radii for ignoring the Kelvin term? What is uncertainty? Line 125. The authors should indicate the uncertainty of aerosol hygroscopicity for 150 nm particles if omitting the Kelvin term.

At line 110, we now cite Wex et al. (2008) and Ruehl et al. (2010) for a discussion on the effects of the Kelvin effect. The remark that the Kelvin effect "is only substantial for very low kappa" was from the perspective of a given growth factor, where lowering kappa implies increasing RH. To be more consistent with conclusions of Wex et al. 2008 and Ruehl et al. 2010, we now state that

"In practice, the Kelvin term is often ignored as its effect is only substantial for RH close to 100% and/or small radii (Wex et al., 2008; Ruehl et al., 2010)"

Furthermore, as suggested we indicate the uncertainty of the growth factor for the example shown in Fig 1 and added the sentence (line 127):

"To further justify neglecting the Kelvin effect in our analysis, we note that for a typical value of κ of 0.15, ignoring the Kelvin effect (Eq. 8) leads to an underestimation of the growth factor of less than 3% at a fw = 0.9."

Line 130. It is not reasonable to classify aerosol as soluble and insoluble particles according to whether the aerosol hygroscopicity parameter (Kappa) is 0. Actually, the kappa of many organics is larger than 0 but they are insoluble. Here can be considered hygroscopic (kappa>0) and non-hygroscopic (kappa=0).

The terminology used in the literature is a bit muddled and the terms soluble/insoluble and hygroscopic/non-hygroscopic or hydrophilic/hydrophobic are often used exchangeable (see for example papers cited in line 133). In line 132, we defined our definition of insoluble as particles with kappa=0. Here, we can refer to Petters & Kreidenweis 2007 who state (referring to a plot of critical supersaturation against dry size) state: "As kappa approaches zero, the particle becomes nonhygroscopic and the slope approaches that expected for an insoluble but wettable particle as predicted by the Kelvin equation, i.e. –1."

Since we already define the terms "insoluble" and "soluble" in line 132, no changes were made to the manuscript related to this comment.

Line 250. More information about RSP is needed, such as the instrument configuration, detection ability, measurement uncertainty, and so on.

Quite some information on RSP is given already in lines 291 and further. About its accuracy we added the sentence (line 293):

"Accuracy of the measured intensity and degree of polarization is generally better than 3% and 0.2%, respectively."

And also (line 300):

"Uncertainties in the derived water fraction, soluble fraction and dry size distributions resulting from measurement uncertainties (see e.g., Knobelspiesse et al., 2012) are expected to be substantially smaller than those arising from the assumptions made in these derivations, as discussed in sections 2, 4 and 5."

Line 435. Whether sulfate is the principal chemical component. The authors can provide the mass fractions of aerosol chemical species measured by AMS.

The mass fractions of organics, ammonium and nitrate are now added to the table S3 in the supplement. We added the following to the text at line 445:

"Mass fractions of ammonium, nitrate and organics are given in table S3. Generally, the majority of the mass other than sulfate is composed of organics."

Figure 3. According to the $\text{kappa}_{ext}$, the hygroscopicity of particles is very weak. If divided by 0.56 using Eq. 28, the kappa of most particles is below 0.1, indicating a large number of organics with weak hygroscopicity. It may be inappropriate for the author to assume that soluble components are represented by sulfate in Line 320.

Daily average kappa values derived from f(RH) are given in table S3 and range from 0.05 to 0.3. Indeed, these kappa values are low because of the large contribution of organics with low hygroscopicity. That is exactly our interpretation. Note that these are single values for bulk aerosol that is generally internally and externally mixed and sampled at 1 Hz. Hence, we do not expect to observe values of kappa representing pure sulfate aerosol for example.

To further clarify this, to line 367 we added:

"Note that the f (80%, 20%) and $\kappa ext$ values represent the hygroscopicity of bulk aerosol observed at 1 Hz."

Figure 5, 6, 9, 11, 12, 14. Some in situ data were used to compare, but the measurement instruments were not introduced in the paper. This information can be added to the supplement. What's the uncertainty of these in situ measurements?

The in situ data are described in quite some detail already in section 3.3. Related to the previous comment, we added the data sampling resolution:

"Data are reported at 1 Hertz resolution."

Uncertainty of the water fraction derived from the in situ probes was also discussed already in lines 332 and further. We added a line with estimated uncertainties of the LAS probe to derived sizes and number concentrations at line 343:

"Uncertainty of the LAS data is estimated at 20%."

Figure 8. The scatter is not well consistent with the model line in (a) and (b). What's the reason?

Possible reasons were already discussed in the paper at line 420 and further:
"The spread in the data may be attributable to natural variation of aerosol dry size distributions during the observation period, as well as uncertainties in effective radius retrievals and water fraction estimates. "

We did not make changes to the paper related to this comment.

Technical advice:

Line 330. "Therefor" should be "Therefore"

Thank you for catching the error. It is corrected.

---

## Referee Report (RR1)

Review:

Thank you for addressing my comments and suggestions, which have all been addressed and incorporated into the revised and improved version of the manuscript. The paper can be accepted for final publication now.